# A Comparison between Three Tuning Strategies for Gaussian Kernels in the Context of Univariate Genomic Prediction

**DOI:** 10.3390/genes13122282

**Published:** 2022-12-03

**Authors:** Osval A. Montesinos-López, Arron H. Carter, David Alejandro Bernal-Sandoval, Bernabe Cano-Paez, Abelardo Montesinos-López, José Crossa

**Affiliations:** 1Facultad de Telemática, Universidad de Colima, Colima 28040, Mexico; 2Department of Crop and Soil Sciences, Washington State University, Pullman, WA 99164, USA; 3Facultad de Ciencias, Universidad Nacional Autónoma de México (UNAM), Mexico City 04510, Mexico; 4Centro Universitario de Ciencias Exactas e Ingenierías (CUCEI), Universidad de Guadalajara, Guadalajara 44430, Mexico; 5International Maize and Wheat Improvement Center (CIMMYT), El Batan, Texcoco 56237, Mexico; 6Hidrociencias, Colegio de Postgraduados, Campus Montecillos, Carretera México-Texcoco Km. 36.5, Montecillo 56230, Mexico

**Keywords:** kernels, tuning, genomic prediction, Gaussian kernel, grid search, Bayesian optimization

## Abstract

Genomic prediction is revolutionizing plant breeding since candidate genotypes can be selected without the need to measure their trait in the field. When a reference population contains both phenotypic and genotypic information, it is trained by a statistical machine learning method that is subsequently used for making predictions of breeding or phenotypic values of candidate genotypes that were only genotyped. Nevertheless, the successful implementation of the genomic selection (GS) methodology depends on many factors. One key factor is the type of statistical machine learning method used since some are unable to capture nonlinear patterns available in the data. While kernel methods are powerful statistical machine learning algorithms that capture complex nonlinear patterns in the data, their successful implementation strongly depends on the careful tuning process of the involved hyperparameters. As such, in this paper we compare three methods of tuning (manual tuning, grid search, and Bayesian optimization) for the Gaussian kernel under a Bayesian best linear unbiased predictor model. We used six real datasets of wheat (Triticum aestivum L.) to compare the three strategies of tuning. We found that if we want to obtain the major benefits of using Gaussian kernels, it is very important to perform a careful tuning process. The best prediction performance was observed when the tuning process was performed with grid search and Bayesian optimization. However, we did not observe relevant differences between the grid search and Bayesian optimization approach. The observed gains in terms of prediction performance were between 2.1% and 27.8% across the six datasets under study.

## 1. Introduction

For breeders to be able to increase genetic gain, they must accurately predict breeding values or phenotypic values. This goal is not so complex when the traits of interest have a simple genetic architecture. However, in traits such as grain yield with a complex genetic architecture that is not well understood, it is more challenging to produce an accurate prediction of breeding values or phenotypic values [1]. For example, in complex trait prediction it is very difficult to accurately model genetic interactions such as epistatic effects, which are common in plant and animal sciences, as well as in biology [2,3,4,5,6], and are difficult to detect and to model efficiently. For this reason, some models had been proposed to capture these complex interaction effects more efficiently and, in this way, quantify the level of influence in understanding the genetic architecture of these traits, thus increasing the accuracy.

One popular method in plant and animal breeding used to model complex interactions more efficiently is the reproducing kernel Hilbert spaces (RKHS) regression [7,8]. The main idea of an RKHS regression is to project the given original input data available in a finite dimensional space onto an infinite dimensional Hilbert space. Kernel methods can be applied with any statistical machine learning algorithm since they transform the input data using a kernel function, which is then used for applying any conventional statistical machine learning method. Empirical evidence suggests that most of the time, better results are obtained with the transformed input. For this reason, RKHS methods are becoming more popular for analyzing nonlinear patterns in datasets collected in plant and animal breeding.

RKHS methods are equally efficient, and popular, for regression and classification problems. One of the most popular methods for classification based on kernels is the support vector machine (SVM) which was proposed to the computer science community in the 1990s by Vapnik [9]. This method has been successful in many fields of study such as speech recognition, text categorization, image recognition, face detection, faulty card detection, junk mail classification, credit rating analysis, and cancer and diabetes classification [10,11].

RKHS methods are also very attractive because in addition to being efficient for capturing nonlinear patterns, they are also efficient for data compression, as the transformed input has less dimensionally than the original input. For example, when the input is a matrix of dimension of order 100 × 100,000, the transformed input has a dimension of order 100 × 100, which has less dimension and as such, can reduce the computational resources required during the training process. There are many transformations (kernel functions) used to capture nonlinear patterns in the original input, and each type of transformation is specialized for capturing some type of nonlinear patterns, but it is impossible to capture all patterns for conventional linear statistical methods.

In the context of genomic selection (GS), RKHS methods are becoming more and more popular since empirical evidence supports that they help increase the prediction accuracy regarding linear methods. For example, Long et al. [12] in a study about body weight of broiler chickens reported a better prediction accuracy of RKHS methods over linear models. Crossa et al. [13] and Cuevas et al. [14] in wheat (Triticum aestivum L.) and maize (Zea mays L.) found that the RKHS methods outperformed the linear methods. However, some authors have also reported minimal differences between RKHS methods and linear models, which is expected when the nonlinear patterns in the data are minimal or non-existing [12,15,16].

However, when an appropriate tuning process is not performed, the prediction performance of RKHS methods regarding linear models is not improved. For example, when the Gaussian kernel is implemented, the bandwidth hyperparameter is set to the median of the average distances or to 1, which in some cases is not optimal and can cause the resulting prediction performance to be equal or worse than conventional linear models. This means that when a nonlinear kernel wants to be implemented, additional work is required to select the best hyperparameters and increase the probability of a better prediction performance. This asseveration is also true for all those machine learning methods that need some hyperparameters to be tuned, since only an optimal tuning process can take the most advantage of the machine learning method at hand. However, it is also true that some default hyperparameters often do a decent job in terms of prediction performance, even though they are not always optimal. For this reason, to obtain all the power of any statistical machine learning method, a careful tuning process should always be performed.

As such, in this paper we carry out a benchmarking study to compare the prediction performance implementing Gaussian kernels in the context of genomic prediction using the univariate Bayesian genomic best linear unbiased predictor (GBLUP) model. 

## 2. Materials and Methods

As the material used was the Bayesian GBLUP model, six real datasets of wheat (datasets 1 to 6 (wheat data) and metrics were used to evaluate the prediction performance. Next, the model used is described, followed by the datasets, and finally, the metrics used to compute the prediction accuracy.

### 2.1. Bayesian GBLUP Model

For prediction performance, the Bayesian GBLUP model implementing Gaussian kernels was used. Since the Gaussian kernel only depends on one hyperparameter, only this hyperparameter was tuned under the following three strategies: (1) no tuning setting to the bandwidth parameter, (2) tuning using the grid search method, and (3) tuning using the Bayesian optimization method.

The model used was
(1)$$Y_{\mathrm{ij}}=\mu +E_{i}+g_{j}+{\mathrm{gE}}_{\mathrm{ij}}+\epsilon_{\mathrm{ij}}$$
where $E_{i}$ are the fixed effects of environments, $g_{j},$ j=1,…,J, are the random effects of lines, $gE_{ij}$ are the random effects of genotype by environment interaction, and $\epsilon_{ij}$ are random error components in the model assumed to be independent normal random variables with mean 0 and variance $\sigma^{2}$. Furthermore, it is assumed that $g={\left(g_{1},\ldots ,g_{J}\right)}^{T}∼N_{J}\left(0,\sigma_{g}^{2}K\right)$, $gL={\left(gE_{11},\ldots ,gE_{1J},\ldots ,gE_{IJ}\right)}^{T}∼N_{IJ}\left[0,\sigma_{gL}^{2}\left(Z_{E}Z_{E}^{T}⨀Z_{g}KZ_{g}^{T}\right)\right]$, where K is the Gaussian kernel (GK) that mimics a covariance matrix to capture the degree of similarity between lines, such as the genomic relationship matrix (Linear kernel) proposed by VanRaden (2008) [17], but is now computed assuming a nonlinear kernel such as the GK, where ⨀ denotes the Hadamard product. The GK was computed using the GK function: (2)$$K\left(x_{i},x_{j}\right)=e^{-\gamma ∥x_{i}-x_{j}{∥}^{2}},\;\mathrm{with}\;\gamma >0$$
where $x_{i}$ and $x_{j}$ are the marker vectors for the ith and jth individuals (genotypes), respectively [18]. It is necessary to point out that the GK function was reparametrized as:(3)$$K\left(x_{i},x_{j}\right)=e^{\log\rho ∥x_{i}-x_{j}{∥}^{2}},\;\mathrm{with}\;\rho \in \left(0,1\right)$$
using the variable change ($\rho =e^{-\gamma }$). Subsequently, the three strategies for tuning the bandwidth (γ) hyperparameter used in this implementation are listed as follows:

(1)Manual tuning (no tuning, denoted as NT) setting, using the value of γ=1, which is equivalent to setting $\rho =e^{-1}$.(2)Tuning using a grid search (GrS) strategy with 26 values in the grid for the values of ρ between 0.01 and 0.999 with increments of 0.04, this means that 26 values of ρ were evaluated. The normalized root mean squared error (NRMSE) was used as metric for choosing the optimal ρ value in the inner testing set. (3)Tuning ρ using the Bayesian optimization (BO) method. The NRMSE was also used as metrics to select the optimal ρ value in the inner testing set.

The implementation of this model with the three strategies of tuning the ρ hyperparameter of the GK was carried out in the R statistical software [19] using the BGLR library [20].

### 2.2. Datasets 1 to 6 (Wheat Data)

Spring wheat lines selected for grain yield analyses from CIMMYT first year yield trials (YT) were used as the training population to predict the quality of lines selected from early year trails (yet) for grain yield analyses in a second year. More details of these datasets can be found in Ibba et al. [21] where the analyses were conducted for 14 traits but in this study, we used only the trait grain yield for each of the six sets of data reported below:

-Dataset 1, Wheat_1 (YT 2013-14/ YT 2014-15), 1,301 lines from the 2013-14 YT and 472 lines from the 2014-2015 EYT trial. In this dataset, only the grain yield trait was used.

-Dataset 2, Wheat_2 (YT2014-15/YT2015-16), 1,337 lines from the 2014-15 YT and 596 lines from the 2015-2016 EYT trial.

-Dataset 3, Wheat_3 (YT2015-16/YT2016-17), 1,161 lines from the 2015-16 YT and 556 lines from the 2016-2017 EYT trial.

-Dataset 4, Wheat_4 (YT2016-17/YT2017-18), 1,372 lines from the 2016-17 YT and 567 lines from the 2017-2018 EYT trial.

-Dataset 5, Wheat_5 (YT2017-18/YT2018-19), 1,386 lines from the 2017-18 YT and 509 lines from the 2018-2019 EYT trial.

-Dataset 6, Wheat_6 (YT 2018-19/YT 2019-20), 1,276 lines from the 2018-19 YT and 124 lines from the 2019-2020 EYT trial. 

All the lines were genotyped using genotyping-by-sequencing (GBS). The TASSEL version 5 GBS pipeline was used to call marker polymorphisms, and a minor allele frequency of 0.01 was assigned for SNP discovery. The resulting 6,075,743 unique tags were aligned to the wheat genome reference sequence (RefSeq v.1.0) (IWGSC 2018) with an alignment rate of 63.98%. After filtering for SNPs with homozygosity >80%, p-value for Fisher’s exact test <0.001 and χ2 value lower than the critical value of 9.2, we obtained 78,606 GBS markers that passed at least one of those filters. These markers were further filtered for less than 50% missing data, greater than a 0.05 minor allele frequency, and less than 5% heterozygosity. Markers with missing data were imputed using the ‘expectation-maximization’ algorithm in the ‘R’ package rrBLUP [22].

### 2.3. Metrics for Evaluation of Prediction Accuracy

In each of the six datasets, the seven outer fold cross validation was implemented [18]. For this reason, 7−1 folds were assigned to the outer-training set and the remaining were assigned to the outer-testing set, until each of the 7 folds were tested once. For tuning the bandwidth, hyperparameter in the Gaussian kernel five nested cross-validations was used, that is, the outer-training was divided into five groups where four were used for the inner training set (80% of the training) and one for the validation (inner-testing) set (20% of the outer training). Next, the average of the five validation folds was reported as the metric of prediction performance to select the optimal hyperparameter (bandwidth of the Gaussian kernel). Then, using this optimal hyperparameter (bandwidth), the GBLUP model was refitted with the whole outer-training set (the 7−1 folds), and finally, the prediction of each outer-testing set was obtained. The mean square error [MSE=$\frac{1}{T}(\overset{T}{\underset{i=1}{\sum }}{\left(y_{i}-\hat{f}\left(x_{i}\right)\right)}^{2}$), with $y_{i}$ denoting the observed value i, while $\hat{f}\left(x_{i}\right)$ represents the predicted value for observation i] and the normalized root mean squared error ($NRMSE=\frac{RMSE}{\bar{y}}$), where $RMSE=\sqrt{\frac{1}{T}(\overset{T}{\underset{i=1}{\sum }}{\left(y_{i}-\hat{f}\left(x_{i}\right)\right)}^{2}}$, was used as a metric to evaluate the prediction accuracy. To compare the three strategies of tuning in terms of MSE, the relative efficiencies were computed between the prediction performance strategy NT and GrS, $$RE_{MSE}=\frac{MSE_{NT}}{MSE_{GrS}}$$
where $MSE_{NT}$ and $MSE_{GrS}$ denote the MSE under the strategy of NT and tuning using GrS, respectively. When $RE_{MSE}>1,$ the best prediction performance in terms of MSE was obtained using the GrS strategy, but when $RE_{MSE}<1,$ the NT strategy was superior in terms of prediction accuracy. When $RE_{MSE}=1,$ both strategies of tuning were equally efficient. We also computed the relative efficiency in terms of MSE between the strategy of NT and tuning using BO strategy and between the GrS and BO strategy. The relative efficiencies just mentioned above were also computed in terms of NRMSE, and the interpretation is the same as in the terms of MSE.

## 3. Results

The results are provided in three main sections, one for each dataset (Datasets 1–3). For each dataset, results are presented based on prediction performance in terms of MSE and NRMSE. Results of datasets 1–3 (wheat 1–3, respectively) are described below. Results from dataset 4–6 (wheat 4–6, respectively) including figures and tables are given in Appendix A. Only three datasets are provided in this section to avoid both redundancy and making the results section unnecessarily long.

### 3.1. Dataset 1 (Wheat 1, YT_13_14 and TY_14_15)

#### 3.1.1. Prediction Performance in Terms of MSE

The MSE for environment YT_13_14 was 0.128, 0.128, and 0.138 for methods BO, GrS, and NT, respectively, and for YT_14_15 environment, the MSE observed was 0.123 (BO), 0.123 (GrS), and 0.133 (NT), while across environments (Global), the observed MSE was 0.125 (BO), 0.125 (GrS), and 0.135 (NT). (Figure 1A and for more details, see Appendix B Table A1.)

The observed relative efficiencies in terms of MSE for the comparison NT/BO were 1.077, 1.088, and 1.079 for environments YT_13_14, YT_14_15 and across environments (Global), respectively. Therefor BO had a better prediction performance than NT in every environment by 7.7% (YT_13_14), 8.8% (YT_14_15) and 7.9% (Global). For the comparison NT/GrS, the relative efficiencies were 1.077 (YT_13_14), 1.081 (YT_14_15), and 1.077 (Global). That is, GrS outperformed NT in every environment by 7.7% (YT_13_14), 8.1% (YT_14_15) and 7.7% (Global). The observed relative efficiencies for the comparison, GrS/BO, were 1.001 (YT_13_14), 1.007 (YT_14_15) and 1.002 (Global). Results showed that both GrS and BO methods had similar performance in every environment. These results can also be observed in Figure 1B and for further details see Table A2.

#### 3.1.2. Prediction Performance in Terms of NRMSE

The NRMSE for environment YT_13_14 was 0.846, 0.846, and 0.877 for methods BO, GrS, and NT, respectively. For environment YT_14_15, the NRMSE was 0.799 (BO), 0.802 (GrS), and 0.837 (NT), while for Global, the NRMSE was 0.460 (BO), 0.461 (GrS), and 0.478 (NT) (see Figure 1C and Appendix B Table A1 for further details). 

The relative efficiencies in terms of NRMSE for comparison NT/BO were 1.037, 1.047, and 1.038 for environments YT_13_14, YT_14_15 and across environments (Global), respectively. Method BO outperformed NT in every single environment by 3.7% (YT_13_14), 4.7% (YT_14_15) and 3.8% (Global). For comparing NT/GrS, the relative efficiencies observed were 1.036 (YT_13_14), 1.043 (YT_14_15), and 1.037 (Global). The GrS method had better performance than the NT method in all three environments by 3.6% (YT_13_14), 4.3% (YT_14_15) and 3.7% (Global). Regarding GrS/BO, the observed efficiencies were 1.000 (YT_13_14), 1.004 (YT_14_15), and 1.001 (Global). Thus, results showed that both GrS and BO had similar prediction accuracy with only very small differences (see Figure 1D and for more details, see Appendix B Table A2).

### 3.2. Dataset 2 (Wheat 2)

#### 3.2.1. Prediction Performance in Terms of MSE

Results on MSE for environment YT_14_15 were 0.072, 0.072, and 0.083 for BO, GrS, and NT, respectively. For the YT_15_16 environment, the MSE was 0.116 (BO), 0.117 (GrS), and 0.149 (NT), while across environments (Global), the MSE was 0.083 (BO), 0.084 (GrS), and 0.101 (NT) (Figure 2A, Appendix C Table A3).

The relative efficiencies for NT/BO were 1.161, 1.278, and 1.211 for environments YT_14_15, YT_15_16 and across environments (Global), respectively. BO had a better prediction performance than NT in every environment by 16.1% (YT_14_15), 27.8% (YT_15_16), and 21.1% (Global). For NT/GrS, the relative efficiencies were 1.156 (YT_13_14), 1.266 (YT_14_15), and 1.205 (Global). Results showed that GrS outperformed NT in every environment by 15.6% (YT_14_15), 26.6% (YT_15_16), and 20.5% (Global). The relative efficiencies for GrS/BO were 1.004 (YT_13_14), 1.009 (YT_14_15), and 1.005 (Global). Both GrS and BO methods had similar performance in every environment (Figure 2B, Table A4).

#### 3.2.2. Prediction Performance in Terms of NRMSE

When comparing the three tuning strategies based on NRMSE, for environment YT_14_15, the NRMSE was 0.930, 0.933, and 1.001 for methods BO, GrS, and NT, respectively, and for YT_15_16 the NRMSE was 0.885 (BO), 0.889 (GrS) and 1.000 (NT). For the environment across traits (Global), the NRMSE was 0.586 (BO), 0.588 (GrS), and 0.645 (NT). (Figure 2C and Appendix C Table A3.)

The comparison of NT/BO for NRMSE was 1.076, 1.130, and 1.100 for environments YT_14_15, YT_15_16 and across environments (Global), respectively. Thus, the BO method outperformed NT in every environment by 7.6% (YT_14_15), 13.0% (YT_15_16), and 10.0% (Global). For NT/GrS, the relative efficiencies were 1.073 (YT_14_15), 1.125 (YT_15_16), and 1.097 (Global). Clearly GrS had better performance than the NT in all three environments by 7.3% (YT_14_15), 12.5% (YT_15_16), and 9.7% (Global). Concerning GrS/BO, the efficiencies were 1.003 (YT_13_14), 1.004 (YT_14_15), and 1.003 (Global). Results showed that both tuning methods GrS and BO had similar prediction accuracy (Figure 2D and Appendix C Table A3).

### 3.3. Dataset 3 (Wheat 3)

#### 3.3.1. Prediction Performance in Terms of MSE

The observed MSE for environment YT_15_16 was 0.062, 0.063, and 0.067 for tuning methods BO, GrS, and NT, respectively. For environment YT_16_17, the MSE for the different tuning methods was 0.199 (BO), 0.197 (GrS), and 0.215 (NT). Across environments (Global), the MSE was 0.103 (BO), 0.102 (GrS), and 0.110 (NT). These results can be observed in Figure 3A and Appendix D Table A5.

The observed relative efficiencies in terms of MSE of the BO, GrS, and NT tuning methods for NT/BO were 1.069, 1.081, and 1.074 for environments YT_15_16, YT_16_17 and Global, respectively. Results showed that BO had a better prediction performance than NT by 6.9% (YT_15_16), 8.1% (YT_16_17), and 7.4% (Global). For the ratio NT/GrS, the relative MSE efficiencies were 1.066 (YT_15_16), 1.088 (YT_16_17), and 1.077 (Global). That is, GrS outperformed NT in every environment by 6.6% (YT_15_16), 8.8% (YT_16_17) and 7.7% (Global). For GrS/BO results were 1.003 (YT_15_16), 0.993 (YT_16_17) and 0.997 (Global). Thus, both tuning methods GrS and BO had similar performance in every environment, as displayed in Figure 3B and shown in Table A6.

#### 3.3.2. Prediction Performance in Terms of NRMSE

Next, we provide the results in terms of NRMSE, where we can observe that for environment YT_15_16, the NRMSE was 0.846, 0.848, and 0.874 for methods BO, GrS, and NT, respectively. While for environment YT_16_17 the NRMSE was 0.864 (BO), 0.861 (GrS) and 0.900 (NT). For across the environment (Global), the NRMSE was 0.522 (BO), 0.521 (GrS), and 0.541 (NT) (Figure 3C and for further details, see Appendix D Table A5).

The observed relative efficiencies in terms of NRMSE in the comparison NT/BO were 1.034, 1.042, and 1.037 for environments YT_15_16, YT_16_17 and across environments (Global), respectively. Clearly BO outperformed NT in every environment by 3.4% (YT_15_16), 4.2% (YT_16_17) and 3.7% (Global). For the NT/GrS ratio, the relative efficiencies in terms of NRMSE were 1.031% (YT_15_16), 1.046% (YT_16_17), and 1.039 (Global). The GrS tuning method had a better prediction accuracy than NT in every environment by 3.1% (YT_15_16), 4.6% (YT_16_17), and 3.9% (Global). When comparing GrS/BO, the relative efficiencies were 1.002 (YT_15_16), 0.996% (YT_16_17), and 0.998% (Global) were very similar (Figure 3D and Appendix D Table A6).

## 4. Discussion

Genomic selection (GS) is revolutionizing plant and animal breeding, as it saves significant resources for selecting the candidate individuals without phenotypic measures of the traits of interest. A statistical machine learning model is trained with a reference population that was genotyped and phenotyped, which is subsequently used to produce predictions of the breeding values or traits of interest used for the selection process [23]. However, the successful implementation of the GS methodology strongly depends on the quality of the predictions. High prediction accuracy requires a successful GS application, but with low prediction accuracies, the outcomes of the GS can be misleading. As such, to improve the accuracy of the outcome of the GS methodology, many factors that impact the performance of the GS methodology need to be optimized.

Since the GS methodology is a predictive methodology, one of the key factors for optimization is the use of the statistical machine learning methods. For example, when the traits to be predicted contain nonlinear patterns, it is obvious that specific statistical machine learning algorithms are required. Kernel methods are powerful tools that capture complex nonlinear patters in the data. For this reason, these methods: (a) are considered promising tools for large-scale and high-dimensional genomic data processing; (b) further improve the scalability of conventional machine learning methods since they can work with heterogeneous inputs; (c) exploit complexity to improve prediction accuracy, but not so much as to increase the understanding of the complexity; (d) contain kernels with great versatility and composite kernels can be built; however, only some can be computed in closed form while others require an iterative process; and (e) kernel methods can be implemented with any statistical machine learning method, which makes these methods really versatile [18]. Nevertheless, the implementation of these methods increases the complexity to obtain a successful implementation since additional hyperparameters need to be tuned; many kernels contain at least one hyperparameter that requires tuning.

However, to guarantee a successful application of kernel methods, it is of paramount importance to choose not only the statistical machine learning algorithm with which the kernels will be implemented, but also carry out a careful tuning process to be able to take the major advantage of its power [24]. The selection of appropriate hyperparameters maximizes model accuracy, considerably reducing the risk of overfitting and producing a model with a variance that is too high. Nevertheless, we need to be aware that the right tuning process is more expensive in terms of computational resources, since hyperparameters are not learned directly through the training process including model parameters. For example, in the grid search approach that was used here to select the optimal hyperparameter, we manually defined a subset of 26 values for the bandwidth hyperparameter space and exhausted all combinations of the specified hyperparameter subsets, that is, each combination performance was evaluated using cross-validation and the best performing hyperparameter combination was chosen. Then, with this optimal hyperparameter combination, the model was refitted to the whole training set, and finally with this model, the predictions were made for the testing set. Since the model is evaluated with each combination of hyperparameters before the final training with the optimal combination of hyperparameters to produce the predictions, the increase in the required computation power is directly proportional to the size of the grid. For this reason, the grid search approach is the most expensive method for hyperparameter tuning, and even the Bayesian optimization is less expensive and more efficient than grid search—even though it requires considerable computation resources. Bayesian optimization is more efficient since it builds a probabilistic model for a given function and analyzes this model to make decisions where to subsequently evaluate the given function. The two main components of Bayesian optimization are: (a) a prior function that captures the main patterns of the unknown objective function and a probabilistic model that describes the data generation mechanism; and (b) an acquisition function (loss function) that describes how optimal a sequence of queries is, usually taking the form of regret [25]. Although these are clear advantages of the Bayesian optimization algorithm, it still requires considerable computational resources. 

In this study, we found that the Bayesian optimization required around only 15% less time for the implementation regarding the grid search approach, and the grid search is approximately 20 times more expensive in computation resources than manual tuning. For this reason, the tuning process, most of the time (when nonlinear patterns are present in the inputs) improves prediction accuracy, although it is significantly more time consuming when it comes to implementation. However, it is important to point out that here the grid search approach was slightly more costly than the Bayesian optimization, since the size of the grid contain 26 values, but if the size of the grid is increased, the grid search is expected to be more expensive in terms of computational resources than the Bayesian optimization. 

Our results show that when using the grid search and Bayesian optimization methods, better prediction performance was obtained regarding the strategy of no tuning. In terms of MSE, we observed gains in prediction accuracy between 7.7 and 8.8% (dataset 1), 15.6 and 27.8% (dataset 2), 6.6 and 8.8% (dataset 3), 15.5 and 16.8% (dataset 4), 4.1 and 10.8% (dataset 5) and 4.5 and 10.8% (dataset 6). On the other hand, in terms of NRMSE, the gains were between 3.6 and 4.7% (dataset 1), 7.3 and 13% (dataset 2), 3.1 and 4.6% (dataset 3), 7.2 and 8.2% (dataset 4), 2.1 and 5.2% (dataset 5), and 2.1 and 4.4% (dataset 6). These results corroborate that an appropriate tuning process is necessary to take advantage of the data and kernel methods. However, we did not find relevant differences between the grid search and Bayesian optimization methods, which in part can be because we used a grid with 26 values for selecting the optimal hyperparameter, which is not a small tuning subset. In addition, the gain in prediction performance between the six datasets was heterogenous, which can be interpreted as the level of nonlinear patterns between the data being are different, because we did not use any criteria to use the six particular datasets.

Finally, with this application we illustrate the important role of the tuning process in Gaussian kernels, and thus are able to take more advantage of kernel methods. This is necessary to guarantee the near or optimal use of kernel methods not only in the context of genomic prediction but in all type of prediction problems. However, the price that we pay for the optimal use of kernel methods is that more computational resources are required to be able to choose the optimal hyperparameters. Moreover, this asseveration applies not only for other types of kernel methods but to all machine learning methods that contain hyperparameters that need to be tuned. Nevertheless, the larger the number of parameters to be tuned, the greater the computational resources are required to obtain the optimal hyperparameters.

## 5. Conclusions

In this research, we compared three tuning strategies (manual, grid search, and Bayesian optimization) using the bandwidth parameter of the Gaussian kernel in the context of genomic prediction under a Bayesian best linear unbiased predictor model (GBLUP). We found that manual tuning produced sub optimal results, when with anticipation, the value of the bandwidth hyperparameter was fixed. However, when the grid search and Bayesian optimization were implemented, we obtained better prediction performance, which means that using these two strategies significantly increases the probability of reaching the optimal hyperparameter values. The superiority of the grid search and Bayesian optimization was observed in the six datasets under study and the gain observed in terms of prediction performance was between 2.1 and 27.8%. However, we did not find a significant difference in the prediction performance between the grid search and Bayesian optimization. In part, this can be attributed to the fact that we used 26 values in the grid search method for the tuning process of the bandwidth hyperparameter. Even though our results are not conclusive, we have empirical evidence corroborating that performing an appropriate tuning process is necessary in order to take advantage of the data and statistical machine learning implemented. 

## Figures and Tables

**Figure 1 genes-13-02282-f001:**
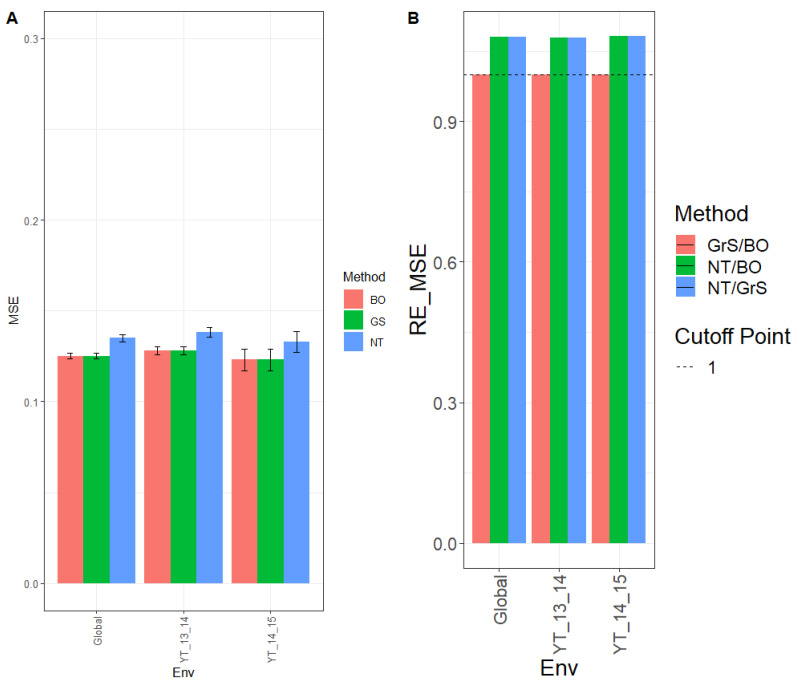
Dataset 1 (Wheat 1). (**A**) Mean squared error (MSE) and the corresponding standard error (SE) of the methods: Bayesian optimization (BO), grid search optimization (GrS), and no tuning (NT) for each environment and across environments of dataset 1. (**B**) Relative efficiency in terms of the mean squared error (RE_MSE) computed by dividing the MSE of NT and BO; NT and GrS; or GrS and BO. Prediction performance is reported for each environment and across environments in the dataset 1. When RE_MSE>1, the denominator method outperforms the numerator in terms of prediction performance. (**C**) Normalized root mean square error (NRMSE) and the corresponding standard error (SE) of the methods Bayesian optimization (BO), grid search optimization (GrS), and no tuning (NT) for each environment and across environments of dataset 1. (**D**) Relative efficiency in terms of the normalized mean squared error (RE_NRMSE) computed by dividing the NRMSE of NT and BO; NT and GrS; or GrS and BO. Prediction performance is reported for each environment and across environments in dataset 1. When RE_NRMSE>1 the denominator method outperforms the numerator in terms of prediction performance.

**Figure 2 genes-13-02282-f002:**
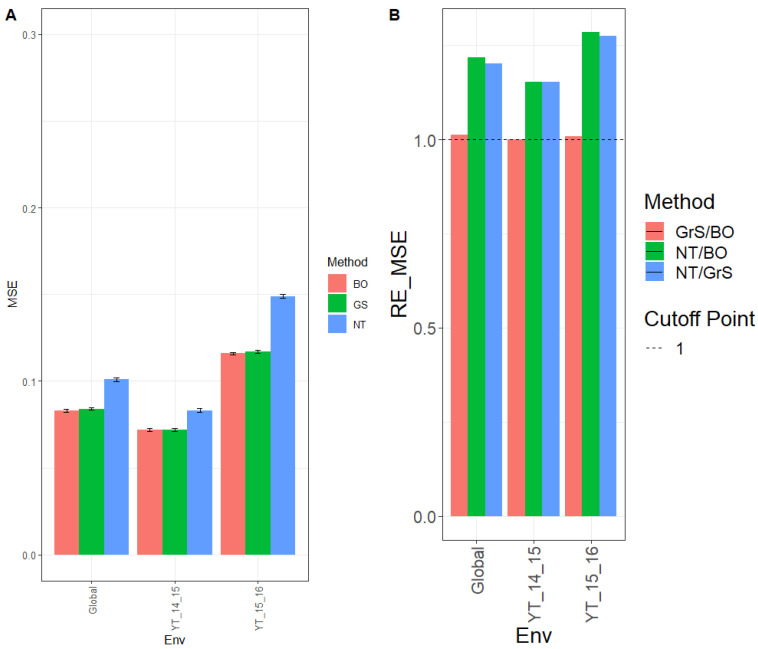
Dataset 2 (Wheat 2). (**A**) Mean squared error (MSE) and the corresponding standard error (SE) of the methods Bayesian optimization (BO), grid search optimization (GrS), and no tuning (NT) for each environment and across environments of dataset 2. (**B**) Relative efficiency in terms of the mean squared error (RE_MSE) computed by dividing the MSE of NT and BO; NT and GrS; or GrS and BO. Prediction performance is reported for each environment and across environments in the dataset 2. When RE_MSE>1 the denominator method outperforms the numerator in terms of prediction performance. (**C**) Normalized root mean squared error (NRMSE) and the corresponding standard error (SE) of the methods Bayesian optimization (BO), grid search optimization (GrS), and no tuning (NT) for each environment and across environments of dataset 2. (**D**) Relative efficiency in terms of the normalized mean squared error (RE_NRMSE) computed by dividing the NRMSE of NT and BO; NT and GrS; or GrS and BO. Prediction performance is reported for each environment and across environments in dataset 2. When RE_NRMSE>1 the denominator method outperforms the numerator in terms of prediction performance.

**Figure 3 genes-13-02282-f003:**
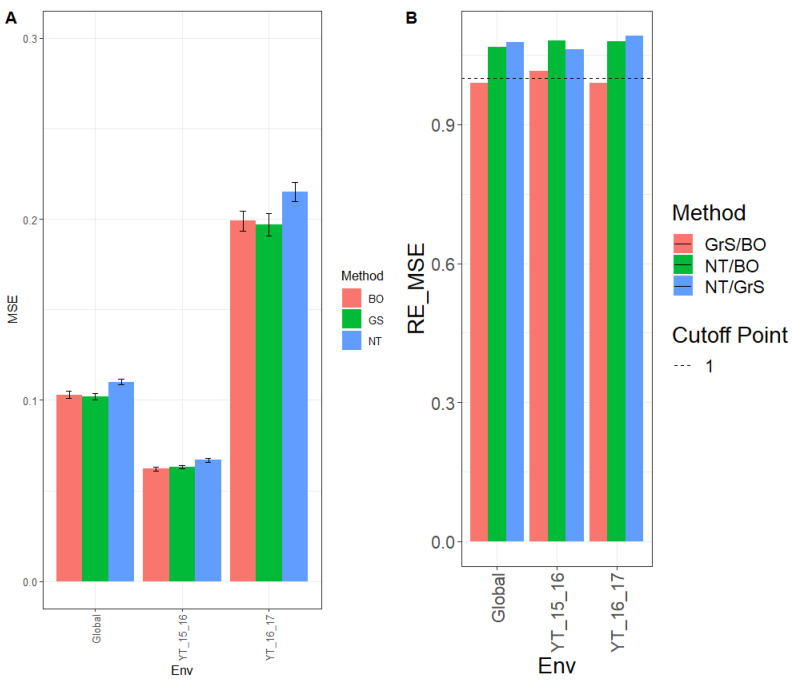
Dataset 3 (Wheat 3) (**A**) Mean squared error (MSE) and the corresponding standard error (SE) of the methods Bayesian optimization (BO), grid search optimization (GrS), and no tuning (NT) for each environment and across environments of dataset 3. (**B**) Relative efficiency in terms of the mean squared error (RE_MSE) computed by dividing the MSE of NT and BO; NT and GrS; or GrS and BO. Prediction performance is reported for each environment and across environments in the dataset 3. When RE_MSE>1 the denominator method outperforms the numerator in terms of prediction performance. (**C**) Normalized root mean square error (NRMSE) and the corresponding standard error (SE) of the methods Bayesian optimization (BO), grid search optimization (GrS), and no tuning (NT) for each environment and across environments of dataset 3. (**D**) Relative efficiency in terms of the normalized mean squared error (RE_NRMSE) computed by dividing the NRMSE of NT and BO; NT and GrS; or GrS and BO. Prediction performance is reported for each environment and across environments in dataset 3. When RE_NRMSE>1 the denominator method outperforms the numerator in terms of prediction performance.

## Data Availability

The genomic and phenotypic of the six wheat datasets (Datasets 1-6) (Wheat 1-6) used in this study can be found at the link https://github.com/osval78/Univariate_Tuning_Kernel_Method.

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
