# Peer review of "A Comparison between Three Tuning Strategies for Gaussian Kernels in the Context of Univariate Genomic Prediction"

_genes, 2022, doi:10.3390/genes13122282_

Round 1

Reviewer 1 Report

#Minor

1) Please do a spelling check for the word "bandwidth." The manuscript contains many parts of the broken text. e.g. band width.

2) Please make figures with the same scale for each category (MSE, RE_MSE, NRMSE, RE_MRMSE). i.e. all MSE figures have the scale between 0-0.3.

3) Please add the number of the cutoff point for all RE_MSE figures (1.00)

#Major

1) Do six real datasets of wheat contain many nonlinear patterns? as the authors describe that the differences between RKHS methods and linear models will be minimal when the nonlinear patterns in the data are minimal or non-existing.

2) Should the computational cost for each tuning strategy be included in the comparison matrix? It will give the audience a better perspective for choosing tuning strategies, as the authors observed no relevant difference between the grid search and Bayesian optimization.

3) Could the authors justify the selection of three main sections (dataset 1-3) to be in the manuscript and put the others (dataset 4-6) in the supplementary materials? Would it be possible to present figures of benchmark results for all datasets in the manuscript? Those figures are crucial for the message of the manuscript, in which the authors use six real datasets to benchmark the tuning strategies.

Author Response

A comparison between three tuning strategies for Gaussian kernels in the context of univariate genomic prediction

Osval A. Montesinos-López1, Arron H. Carter2, David Alejandro Bernal Sandoval1, Bernabe Cano-Paez3, Abelardo Montesinos-López4,* and José Crossa5,6,*

RESPONSE TO REVIEWER 1

RESPONSE: the authors recognize the valuable time invested by the reviewer in reading this research. These days being a reviewer is indeed a great work for the advance of science. Thank you very much!

#Minor

  • Please do a spelling check for the word "bandwidth." The manuscript contains many parts of the broken text. e.g. band width.

RESPONSE: Thanks. Correction done in the new version of the paper. See lines: 86, 169. 173, 174

  • Please make figures with the same scale for each category (MSE, RE_MSE, NRMSE, RE_MRMSE). i.e. all MSE figures have the scale between 0-0.3.

RESPONSE: Correction done in the new version of the paper. See figures 1, 2 and 3, and those in supplemental materials.

  • Please add the number of the cutoff point for all RE_MSE figures (1.00)

RESPONSE: Many thanks. Correction done in the new version of the paper.  See figures 1, 2 and 3, and those in supplemental materials.

#Major

  • Do six real datasets of wheat contain many nonlinear patterns? as the authors describe that the differences between RKHS methods and linear models will be minimal when the nonlinear patterns in the data are minimal or non-existing.

RESPONSE: We only selected this data sets, but without any criteria of degree of non-linear patterns, for this reason we can speculate that for those data sets in which a larger gain in prediction performance was obtained is due to the fact that they contain more linear patters. See lines: 403-406.

  • Should the computational cost for each tuning strategy be included in the comparison matrix? It will give the audience a better perspective for choosing tuning strategies, as the authors observed no relevant difference between the grid search and Bayesian optimization.

RESPONSE: Correction done in the new version of the paper. See lines: 381-391.

3) Could the authors justify the selection of three main sections (dataset 1-3) to be in the manuscript and put the others (dataset 4-6) in the supplementary materials? Would it be possible to present figures of benchmark results for all datasets in the manuscript? Those figures are crucial for the message of the manuscript, in which the authors use six real datasets to benchmark the tuning strategies.

RESPONSE: We thank the reviewer and understand the main point. However, only three data sets are provided in this section to avoid redundancy and making so long the results section. However, also in the discussion is show the average performance of data sets 4-6, and we say that the gain in prediction performance between the six data set was heterogenous which can be interpreted as that the level of non-linear patterns between the data sets are different, because we not use any particular criteria for selecting the used six data sets.  See lines: 197-198; 393-398; 403-406.

Reviewer 2 Report

Comments and Suggestions for Authors

The manuscript submitted for review is focused on the comparing of three tuning strategies (Manual, grid search and Bayesian optimization) in the context of genomic prediction under a Bayesian best linear unbiased predictor model, being the genomic prediction a revolutionizing step in plant breeding since candidate genotypes can be selected without the need to measure their trait in the field. The Reproducing Kernel Hilbert Spaces (RKHS) regression is becoming more and more popular method in plant and animal breeding increasing the prediction accuracy of breeding or phenotypic values of candidate genotypes. For the study of the prediction performance the univariate Bayesian genomic best linear unbiased predictor (GBLUP) model was applied on six real data sets of wheat (Triticum aestivum L.). Was found that the careful performation of tuning process is of high importance for major benefits. The best prediction performance was observed when the tuning process was performed with grid search and Bayesian optimization.

The manuscript is structured according to the requirements to authors of “Genes”. The methods applied are appropriates, clearly described and sufficient. The illustrative material (tables and figures) are representative and of good quality.

Some comments need to be made:

Introduction

-          When authors are cited in the text, only the number of the cited literary source is given, without the year: Vapnik (1995) [9], Long et al. (2010) [12], Cuevas et al. (2019) [14], etc., must be: Vapnik [9], Long et al. [12], Cuevas et al. [14]...

-          The last part of this paragraph starting from “Since Gaussian kernel only depends on one hyperparameter…” have to be moved in “Materials and Methods”

Materials and Methods

-          This paragraph should begin with a description of the material used: “As material six real data sets of wheat were used:…. (the information is given below - Datasets 1 to 6 (Wheat data) from line 131)

-          After that follows the description of the model:

Bayesian GBLUP model

For prediction performance the Bayesian GBLUP model implementing Gaussian kernels was used. Since the Gaussian kernel only depends on one hyperparameter, only this hyperparameter was tuned under the following three strategies: (1) no tuning setting to one the band width 102 parameter, (2) tuning using the grid search method and (3) tuning using the Bayesian optimization method.

The model used was ….

Results

-          There are not commented the results obtained for Datasets 4 to 6, and on what basis is decided the figures representing the results for Datasets 1 to 3 to be presented in the main manuscript and these representing the results for Datasets 4 to 6 – in Supplementary materials.

Author Response

A comparison between three tuning strategies for Gaussian kernels in the context of univariate genomic prediction

Osval A. Montesinos-López1, Arron H. Carter2, David Alejandro Bernal Sandoval1, Bernabe Cano-Paez3, Abelardo Montesinos-López4,* and José Crossa5,6,*

REVIEWER 2 Report Form Top of Form

Comments and Suggestions for Authors

The manuscript submitted for review is focused on the comparing of three tuning strategies (Manual, grid search and Bayesian optimization) in the context of genomic prediction under a Bayesian best linear unbiased predictor model, being the genomic prediction a revolutionizing step in plant breeding since candidate genotypes can be selected without the need to measure their trait in the field. The Reproducing Kernel Hilbert Spaces (RKHS) regression is becoming more and more popular method in plant and animal breeding increasing the prediction accuracy of breeding or phenotypic values of candidate genotypes. For the study of the prediction performance the univariate Bayesian genomic best linear unbiased predictor (GBLUP) model was applied on six real data sets of wheat (Triticum aestivum L.). Was found that the careful performation of tuning process is of high importance for major benefits. The best prediction performance was observed when the tuning process was performed with grid search and Bayesian optimization.

The manuscript is structured according to the requirements to authors of “Genes”. The methods applied are appropriates, clearly described and sufficient. The illustrative material (tables and figures) are representative and of good quality.

Some comments need to be made:

 RESPONSE: Many thanks for your time invested revising this manuscript. Authors, highly appreciated it

Introduction

-          When authors are cited in the text, only the number of the cited literary source is given, without the year: Vapnik (1995) [9], Long et al. (2010) [12], Cuevas et al. (2019) [14], etc., must be: Vapnik [9], Long et al. [12], Cuevas et al. [14]...

RESPONSE: Yes, indeed you are right and sorry for the trouble. Correction done in the new version of the paper. See lines: 78-80 and 140.

-          The last part of this paragraph starting from “Since Gaussian kernel only depends on one hyperparameter…” have to be moved in “Materials and Methods”

RESPONSE: Correction done in the new version of the paper. Se lines 109-112.

Materials and Methods

-          This paragraph should begin with a description of the material used: “As material six real data sets of wheat were used:…. (the information is given below - Datasets 1 to 6 (Wheat data) from line 131)

RESPONSE: Yes and thanks. See lines: 102-105.

-          After that follows the description of the model:

Bayesian GBLUP model

For prediction performance the Bayesian GBLUP model implementing Gaussian kernels was used. Since the Gaussian kernel only depends on one hyperparameter, only this hyperparameter was tuned under the following three strategies: (1) no tuning setting to one the band width 102 parameter, (2) tuning using the grid search method and (3) tuning using the Bayesian optimization method.

The model used was ….

RESPONSE: Correction done in the new version of the paper. See lines 108-112.

Results

-          There are not commented the results obtained for Datasets 4 to 6, and on what basis is decided the figures representing the results for Datasets 1 to 3 to be presented in the main manuscript and these representing the results for Datasets 4 to 6 – in Supplementary materials.

RESPONSE: As we responded to Reviewer 1 we understand the reviewer concern. Our mail point was that by providing results on only three data sets we avoid redundancy and making so long the results section. However, also in the discussion is show the average performance of data sets 4-6, and we say that the gain in prediction performance between the six data set was heterogenous which can be interpreted as that the level of non-linear patterns between the data sets are different, because we not use any particular criteria for selecting the used six data sets.  See lines: 197-198; 393-398; 403-406.

Round 2

Reviewer 1 Report

-